# Sleep-Disordered Breathing in Patients with Chronic Heart Failure and Its Implications on Real-Time Hemodynamic Regulation, Baroreceptor Reflex Sensitivity, and Survival

**DOI:** 10.3390/jcm13237219

**Published:** 2024-11-27

**Authors:** Anna S. Lang-Stöberl, Hannah Fabikan, Maria Ruis, Sherwin Asadi, Julie Krainer, Oliver Illini, Arschang Valipour

**Affiliations:** 1Karl Landsteiner Institute of Lung Research and Pulmonary Oncology, Clinic Floridsdorf, 1210 Vienna, Austria; hannah.fabikan@extern.gesundheitsverbund.at (H.F.); julie.krainer@extern.gesundheitsverbund.at (J.K.);; 26th Department of Internal Medicine with Pulmonology, Clinic Hietzing, Vienna Healthcare Group, 1130 Vienna, Austria; 3Department of Pediatrics, Clinic Donaustadt, Vienna Healthcare Group, 1220 Vienna, Austria; 4Department of Respiratory and Critical Care Medicine, Clinic Floridsdorf, Vienna Healthcare Group, 1210 Vienna, Austria

**Keywords:** sleep-disordered breathing, obstructive sleep apnea, Cheyne–Stokes breathing, chronic heart failure, baroreceptor reflex sensitivity

## Abstract

**Background:** Impairment in autonomic activity is a prognostic marker in patients with heart failure (HF), and its involvement has been suggested in cardiovascular complications of obstructive sleep apnea syndrome (OSAS) and Cheyne–Stokes respiration (CSR). This prospective observational study aims to investigate the implications of sleep-disordered breathing (SDB) on hemodynamic regulation and autonomic activity in chronic HF patients. **Methods:** Chronic HF patients, providing confirmation of reduced ejection fraction (≤35%), underwent polysomnography, real-time hemodynamic, heart rate variability (HRV), and baroreceptor reflex sensitivity (BRS) assessments using the Task Force Monitor. BRS was assessed using the sequencing method during resting conditions and stress testing. **Results:** Our study population (*n* = 58) was predominantly male (41 vs. 17), with a median age of 61 (±11) yrs and a median BMI of 30 (±5) kg/m^2^. Patients diagnosed with CSR were 13.8% (8/58) and 50.0% (29/58) with OSAS. No differences in the real-time assessment of hemodynamic regulation, heart rate variability, or baroreceptor reflex function were found between patients with OSAS, CSR, and patients without SDB. A subgroup analysis of BRS and HRV in patients with severe SDB (AHI > 30/h) and without SDB (AHI < 5) revealed numerically reduced BRS and increased LF/HF-RRI values under resting conditions, as well as during mental testing in patients with severe SDB. Patients with moderate-to-severe SDB had a shorter overall survival, which was, however, dependent upon age. **Conclusions:** Chronic HF patients with severe SDB may exhibit lower baroreceptor function and impaired cardiovascular autonomic function in comparison with HF patients without SDB.

## 1. Introduction

Heart failure (HF) is a multifactorial and life-threatening clinical syndrome defined by a combination of cardiac abnormalities, elevated natriuretic peptide levels, and pulmonary or systemic congestion, and classified based on the left ventricular ejection fraction (LVEF) [1,2]. With a prevalence rate of 1.7% across Europe, it is a major cause of mortality, morbidity, and poor quality of life [1,3]. The most important factor that determines the outcome of patients with HF is its constant progression [4]. Optimized pharmacological treatment and cardiac resynchronization therapy have improved treatment, however, treatment management is often complicated due to multiple concomitant diseases [5,6]. Close interrelations and potentiation of HF and concomitant diseases exist, which is why the identification of these diseases and their contributing features to HF progression and prognostic impairment is critical [5]. Sleep-disordered breathing (SDB) has been shown to have much higher prevalence rates in HF patients, which range from 46% up to 82%, compared to the general population [7,8,9,10]. It has been suggested that similarities in pathophysiology between HF and SDB are linking the two conditions [7]. The two major types of SDB are obstructive sleep apnea (OSA) and central sleep apnea (CSA). While OSA is characterized by a pharyngeal collapse during sleep resulting in upper airway obstruction, CSA is characterized by a periodic reduction or cessation of respiratory effort resulting in inappropriate hyperventilation, both causing apnea or hypopnea [11,12]. Particular for CSA is Cheyne–Stokes respiration (CSR), which is a periodic pattern of hypoventilation following hyperventilation [13]. In patients with HF, both OSA and CSR are common, at 36% and 40%, respectively, and have been associated with an increased mortality [7,14,15]. Reasons for this excess mortality, however, are yet incompletely understood [16]. The combined effects of intermittent hypoxia, arousals from sleep, negative intrathoracic pressure during obstructive apneas, and interference with neuro-humoral systems by increasing sympathetic and renin–angiotensin–aldosterone activity may worsen heart failure in HF patients with SDB [7,14,15,17]. The hemodynamic aftereffects of negative intrathoracic pressure during obstructive apnea include significant reductions in cardiac indices in patients with HF, while central apneic events show significant increases [16,18]. Markers that reflect the autonomic activity are heart rate variability (HRV) and baroreflex sensitivity (BRS), which is a regulatory mechanism crucial in the homeostasis of blood pressure by influencing the peripheral vascular tone and cardiac output [19]. Both reductions in HRV and BRS have been shown to predict death and increase cardiovascular morbidity and mortality in HF patients, respectively [20,21,22]. Consequently, the aim of this study was to analyze the implications of SDB on real-time hemodynamic regulations, HRV, and BRS in chronic HF patients.

## 2. Materials and Methods

### 2.1. Study Design and Participants

This study was conducted as a prospective observational study to investigate the implications of SDB on hemodynamic regulations, HRV, and BRS in chronic HF patients. The study was approved by the Ethics Committee of the City of Vienna (EK 08-058-0508), and all participants gave written informed consent before study participation. The study was conducted in accordance with the Declaration of Helsinki.

During April 2008 and April 2012, patients with stable chronic HF aged from 18 to 80 years were recruited from five cardiologic departments of the Vienna Healthcare Association during ambulatory visits. Patients were eligible for inclusion if they (1) had stable chronic HF (NYHA I-IV) for at least 6 months, (2) had been under maximized therapy without changes of symptoms or treatment for at least 8 weeks prior to study entry, and (3) have had no hospitalizations for at least 3 months prior to study entry. (4) Additionally, patients had to provide an echocardiography not older than 6 months confirming a reduced ejection fraction of ≤35%. Patients were excluded from study participation if they suffered from unstable angina, had had a myocardial infarction, suffered a stroke within the previous 12 months, had primary pulmonary hypertension, a congenital heart defect, a primary valvular heart disease, abuse of benzodiazepine, opiate, alcohol or cocaine, severe kidney (serum creatinine level > 3 mg/dL) or liver disease (GPT > 3× ULN), clinically relevant CNS diseases, known moderate-to-severe chronic obstructive pulmonary disease (FEV1/(F)VC < 70% and FEV1 < 50%) or known restrictive ventilatory impairment with TLC < 70%, untreated hormonal diseases, implantation of a pacemaker or defibrillator within the previous 6 months, or an aortocoronary bypass surgery or lung resection within the previous 12 months.

### 2.2. Measurements

All chronic HF patients eligible for study participation were referred from different cardiologic departments in Vienna to the sleep laboratory of the Otto Wagner Hospital, Vienna, Austria. They were submitted to a detailed medical history including medication, categorized according to NYHA, and assessed for cardiovascular risk factors. Additionally, the patients completed questionnaires regarding sleep quality (Pittsburgh Sleep Quality Index (PSQI)), daytime sleepiness (Epworth Sleepiness Scale (ESS)), and living with heart failure (Minnesota Living with Heart Failure Questionnaire (MLHFQ)). The PSQI graded patients with a score > 5 as bad sleepers [23]. Excessive daytime sleepiness was considered with an ESS > 10 [24]. The MLHFQ classified patients’ health-related quality of life as good (<24), moderate (24–45), and poor (>45) [25]. Treatment for SDB and survival of patients was followed up using medical records in February 2024.

#### 2.2.1. Polysomnography

All patients completed a polysomnography (PSG) at the sleep laboratory. PSGs were scored according to the scoring criteria of the American Academy of Sleep Medicine (AASM) 2007. CSR was diagnosed with an AHI ≥ 5/h of central apneas and/or central hypopneas associated with at least 3 consecutive cycles of crescendo/decrescendo breathing patterns or as episodes of at least 10 consecutive minutes of crescendo and decrescendo changes in breathing amplitude. OSAS was diagnosed with an obstructive apnea–hypopnea index (AHI) ≥ 15/h or ≥5/h plus symptoms suggestive of SDB [26]. All sleep studies were reviewed manually by trained personnel and checked by clinical sleep experts.

#### 2.2.2. Real-Time Hemodynamic Measurements and Baroreceptor Reflex Sensitivity

The morning after the PSG, the patients underwent an arterial blood gas analysis and completed a noninvasive measurement of hemodynamics, HRV, and BRS, which was conducted using the “Task Force Monitor” (CNSystems, Graz, Austria). The Task Force Monitor combines impedance cardiography with a continuous measurement of non-invasive arterial pressure. Measurements were performed in a quiet and dimmed room with the patient resting in a supine position. First, baseline measurements were obtained at rest, followed by stimulation by mental arithmetic testing.

The monitoring of hemodynamic functions was conducted using transthoracic impedance cardiography, 3-channel ECG, oscillometric and continuous beat-to-beat measurements of the arterial blood pressure, and pulse oximetry by the Task Force Monitor. Systolic and diastolic beat-to-beat arterial blood pressure was obtained by the vascular-unloading technique of the finger and corrected by oscillometric blood pressure measurements at the brachial artery of the contralateral arm. Real-time beat-to-beat stroke volume (SV), stroke index (SI), cardiac output (CO), cardiac index (CI), and total peripheral resistance (TRP) were obtained by utilizing an improved method of transthoracic impedance cardiography [27]. HRV was calculated by frequency domain analysis, by which the frequencies of continuously measured ECG recordings are analyzed. Divided into very low frequency, low frequency, and high frequency, vagal and sympathetic modulations can be reflected. The ratio between low and high frequency (LF/HF) represents the sympathovagal balance [28].

BRS was evaluated by the Task Force Monitor, which has previously been validated [27,29,30] and used in various patient groups, including healthy subjects undergoing artificially induced hemodynamic changes [31,32], patients with COPD [33], and patients with different types of cardiovascular disease, including those with heart failure [34]. To account for the reverse relationship between HR and BRS, the HR at rest was used as an independent variable in a multivariable regression analysis for adjustment [35].

### 2.3. Statistical Methods

All data were analyzed with Stata version 14 (StataCorp LP, College Station, TX, USA). The results are shown as the mean (SD) or median and quartiles or percentages as appropriate. The distribution of continuous variables was assessed using skew statistics and normal quantile plots. Differences between patients with OSAS, CSR, or without SDB were assessed using a one-way ANOVA or the Kruskal–Wallis test as appropriate. Differences between the two groups of patients were investigated using an unpaired *t*-test or the Kruskal–Wallis test as appropriate. The individual association between SDB and each potential risk factor was assessed using a univariate logistic regression analysis. For associations between BRS slope mean and potential risk factors, initially, a univariate logistic regression analysis was applied, followed by a multiple logistic regression analysis for heart rate correction.

Overall survival (OS) was defined as the time interval from the date of SDB diagnosis to the date of death independent of cause. Median OS was calculated using the Kaplan–Meier estimator and a confidence interval (CI) of 95%. Median follow-up was calculated using the reversed Kaplan–Meier estimator. Univariate potential prognostic predictors for survival were assessed using the log-rank test with a level of significance of 5% (chi square *p* = 0.05), followed by stepwise forward multivariate Cox regression analysis. A two-sided *p*-value < 0.05 was considered statistically significant. All methods and results are reported according to the Strengthening the Reporting of Observational Studies in Epidemiology (STROBE) guidelines [36].

## 3. Results

### 3.1. Baseline Characteristics

Of the 96 patients eligible for study participation, 93 patients could be included in the study. Thirty-one patients dropped out, as they were either unable to complete the polysomnography or the hemodynamic measurements. Thus the data of 62 patients were analyzed. The patients were diagnosed according to the scoring criteria described in the method Section 2.2.1. Four patients were diagnosed with mixed SDB and were therefore excluded. The final analysis was conducted with 58 patients. The study flowchart is shown in Figure 1.

Classification of chronic HF was ischemic in 22 patients (37.9%) and non-ischemic in 36 patients (62.1%). The total study population included more men than women (41 vs. 17), with a median age of 61(±11) years and a median BMI of 30 (±5) kg/m^2^. Twenty (34.5%) patients were in NYHA class I, 33 (56.9%) in NYHA class II, and 5 (8.6%) in NYHA class III. Comorbidities were common among the patients. Thirty-five patients (60.3%) had been diagnosed with hypertension, 20 (34.5%) with diabetes mellitus, and 26 (44.8%) with hyperlipidemia. Regarding heart failure medication, 94.8% of patients received betablockers, 79.3% diuretics, 73.7% ACE inhibitors, 31.6% ATII blockers, and 13.8% digitalis. Ten patients had an ESS > 10, demonstrating a prevalence of excessive daytime sleepiness (EDS) of 17.2% in the overall study population. According to the results of the MLHFQ, health-related quality of life was assessed in 51 patients. The quality of life was categorized as good in 30 patients (58.8%), moderate in 14 patients (27.5%) and poor in 7 (13.7%). Thirty-three patients (58.9%) were poor sleepers according to the PSQI assessed in 57 patients.

### 3.2. Prevalence of SDB

SDB was found to be highly prevalent, with 63.8% in this cohort of chronic HF patients (Figure 2A). Of all 58 patients, 8 (13.8%) were diagnosed with CSR and 29 (50.0%) with OSAS. The severity of SDB based on AHI was that 19 patients (32.8%) had mild SDB (AHI 5 ≤ AHI > 15), 7 patients (12.1%) had moderate SDB (15 ≤ AHI > 30), and 12 patients (20.7%) had severe SDB (AHI < 30). Severe SDB was prevalent in five patients (17.2%) diagnosed with OSAS and in seven patients (87.5%) diagnosed with CSR (Figure 2B). The results of the overnight sleep examination are displayed in Table 1. There were significant differences in AHI (*p* = 0.000) (Figure 3), REM-AHI (*p* = 0.009), supine AHI (*p* = 0.000), arousals (*p* = 0.001), minimal SaO_2_ (*p* = 0.000), mean SaO_2_ (*p* = 0.003), and paCO_2_ (*p* = 0.038) between groups. The clinical characteristics of patients classified into OSAS, CSR, and those without SDB are shown in Table 2. We found significant differences regarding the smoking status between groups (*p* = 0.040).

### 3.3. Real-Time Hemodynamic Measurements and Heart Rate Variability in Chronic HF Patients with and Without SDB

HF patients with OSAS, CSR, and those without SDB did not show any significant between-group differences regarding hemodynamic measurements or heart rate variability at rest or under mental testing conditions (Table 3). However, an analysis of the patients with severe SDB and without SDB showed numerical differences in the LF/HF ratio and LF/HF-RRI during rest and under stress conditions (Table 4).

### 3.4. Baroreceptor Reflex Sensitivity

While there was a trend towards a numerically lower mean baroreceptor slope in patients with SDB, there were, overall, no significant between-group differences regarding parameters of baroreceptor function in heart failure patients with or without SDB (Table 5). Patients with severe SDB (*n* = 12) compared to patients without SDB (*n* = 21) had numerically reduced BRS values under resting conditions, as well as during mental testing (Figure 4). Multivariable regression analysis showed no correlation among AHI, REMAHI, supine AHI, arousals, min SaO_2_, Mean SaO_2_, or paCO_2_ with either BRS under rest or under testing conditions independent of heart rate.

### 3.5. Treatment in Patients with SDB

Of the 37 patients diagnosed with SDB, 37.8% (14/37) received positive airway pressure therapy. Of these 14 patients, two patients had mild, two patients had moderate, and ten patients had severe SDB. Lateral positioning training was received by 35.1% (13/37), of which six patients were additionally advised to reduce weight and one was prescribed nightly O2 therapy by O2 concentration. Two patients were treated with nightly O2 therapy by O2 concentration alone. Eight patients received no therapy, six of them due to a low disease burden, while two patients with severe OSA rejected therapy.

### 3.6. Survival Analysis

The median follow-up time was 14.4 (14.0; 15.7) years. The overall survival at the end of the study period was 43.1%. The median survival of CHF patients with OSAS and with CSR was similar, with 11.4 (6.2; NA) and 12.1 (4.2; NA) years, respectively. However, patients with no or mild SDB and patients with moderate or severe SDB showed significant differences in the log-rank test for equality of survivor functions (log-rank *p* = 0.047; HR= 1.99 (0.10–3.98) *p* = 0.051) (Figure 5). Patients with moderate-to-severe SDB had a median survival of 7.9 (5.0; 13.1) years, while the median survival for patients with no or mild SDB was not reached. After adjustment for age, the association did not remain significant (HR= 1.31 (0.63–2.69) *p* = 0.456). An analysis with a stepwise forward Cox regression model showed no significance regarding survival for any of the assessed hemodynamic measurements or positive airway pressure therapy.

## 4. Discussion

This study provides comprehensive hemodynamic measurements and assessment of parameters of autonomous cardiovascular control in patients with chronic HF, with and without SDB. While we observed no differences in hemodynamic regulation or baroreceptor function between chronic HF patients with OSAS, CSR, or without SDB, we report a trend towards numerically lower mean baroreceptor function in patients with OSAS and CSR. A subgroup analysis of chronic HF patients with severe SDB compared to patients without SDB showed a trend for a decrease in BRS and an increase in HRV in patients with severe SDB. The survival data showed a reduced survival for chronic HF patients with moderate-to-severe SDB compared to chronic HF patients with no or mild SDB, which was, however, dependent on age. We found that neither hemodynamic measurements nor therapy with positive airway pressure influenced overall survival.

In the present cohort, 63.8% of patients with stable chronic HF suffered from SDB. Although the data on SDB prevalence rates in chronic HF patients range from 46% to 82% [8,10], the observed high prevalence of SDB in our study is in accordance with large studies that included more than 100 patients [7,37,38]. Oldenburg et al. screened 700 patients with chronic HF and found SDB to be present in 76% of patients, with similar patient characteristics to our study regarding patients’ age, BMI, and male predominance [7]. Moreover, studies by Sin et al. and Schulz et al. reported a prevalence of 61% in 450 patients and 71% in 203 patients with chronic HF, respectively [37,38]. Conversely, a much lower prevalence was demonstrated by a large multicenter study from the German Schlaf registry, reporting a prevalence of 46% in a sample of 6876 patients [8]. The lower prevalence of SDB in this study may be explained by the fact that SDB screening was conducted using the ApneaLink instead of the gold-standard PSG, resulting in a potential underestimation of the AHI, as shown in a validation study in a subset of patients of the Schlaf registry [8]. In contrast, a very high prevalence rate of 82% was found by Kishan et al. in the assessment of SDB in the Indian population with predominantly ischemic HF. This high rate, however, may be due to a possible selection bias, as only hospitalized patients were included in the study [10]. In our study, the predominant type of SDB was OSAS with 50.0%, while 13.8% of patients suffered from CSR. Similarly, predominant percentages of OSA have been reported by Schulz et al. (OSA 43%, CSR 28%), Herrscher et al. (53% OSA, 27% CSR), Paulino et al. (57% OSA, 24%CSR), and Kishan et al. (OSA 59%, 22% CSR) [10,38,39,40]. However, similar proportions of CSR and OSA [7,37] as well as a higher prevalence of CSR over OSA [41,42] have also been reported by various other studies. The relatively low prevalence of CSR in our study may be due to the fact that most patients (94.8%) in our report were on maximized pharmacological therapy, which has been shown to reduce CSR [43].

Both OSA and CSR in patients with HF are associated with increased mortality [14,15]. The combined effects of intermittent hypoxia, frequent arousals, and autonomic dysfunction in SDB have been suggested to contribute to increased mortality in HF patients [16]. While both patients with OSA and CSR exhibit increased HR and a high BP, SV and CO are diametrically contrasting in both diseases [16,18,44,45]. Hyperventilation—a key component in CSR—is reported to be associated with an increased CO in both healthy subjects as well as in HF patients, even in patients receiving therapy with beta-blocking agents [18]. Upper-airway occlusion during obstructive apneic events, on the other hand, results in inspiratory negative intrathoracic pressure increasing right ventricular preload, while hypoxia causes pulmonary vasoconstriction increasing the left ventricular afterload, leading to reductions in both SV and CO [16]. In patients with HF, a comparison of obstructive and central sleep apnea found increases in SV and CO during central apnea–hypopnea, while decreasing during obstructive apnea–hypopneas. These findings indicate that obstructive and central respiratory events have opposite effects on SV and CO [16]. Subsequently, in HF patients, hyperventilation in CSR has even been suggested to be compensatory for a failing heart [18]. In our study, however, we found no significant differences between SV or CO in HF patients with OSAS, CSR, or without SDB. Compared to the hemodynamic effects within a similar cohort of CSR patients likewise assessed with the Task Force Monitor, our CSR patients showed a lower heart rate (63 (56–70) vs. 75 (±17) bpm) and a higher stroke volume (74 (±16) vs. 64 (±29) mL) but a similar cardiac output (4.7 (±0.7) vs. 4.4 (±1.3) L/min) [18]. Nevertheless, a study assessing a slightly younger HF cohort with OSA, CSR, and without SDB reported an almost identical HR to ours [46]. HRV, characterized as the variability of the beat-to-beat intervals of the heart, is known to be abnormal in HF patients [21]. An analysis of the frequency domain measures of HRV assessing the LF/HF ratio offers reliable information on autonomic activity, as an increased LF/HF ratio reflects sympathetic predominance and blunted parasympathetic activity [20,21,46]. In HF patients with SDB, a shift towards sympathetic predominance has been reported [46]. In our study, we observed numerically higher LF/HF ratios in patients with OSAS and CSR than in patients with no SDB. However, these differences did not reach significance. These findings are in accordance with the study of Szollosi et al. that assessed various domain measures of HRV in patients with HF and also found only numerical increases in the LF/HF ratio between OSA and CSR [46]. It has to be pointed out that, while in patients with HF reductions of HRV are reported and associated with a poor outcome, CSR—likewise associated with a poor outcome in HF patients—increases HRV [15,20,21,46]. Therefore, it has been proposed that, while CSR and OSA increase HRV during periods of abnormal breathing, HRV may be reduced during waking hours with stable breathing [46]. In patients with OSA, special attention has been given to rapid eye movement (REM) sleep because it has been suggested that upper airway collapsibility increases during REM sleep due to muscle hypotonia, resulting in longer obstructive apneic events with greater oxygen desaturation [47]. Furthermore, as sympathetic activity has been associated with REM sleep in healthy subjects, it has been suggested that OSA during REM sleep may have a significant effect on cardiovascular risk and autonomic dysfunction [48]. However, studies analyzing patients with REM-associated OSA could not confirm an increased autonomic dysfunction during REM sleep. Especially, since REM sleep only accounts for approximately 20% of total sleep time, the overall effect on autonomic dysfunction was found to more depend on the overall OSA severity [48]. Although we could not assess REM-associated OSA within our study cohort due to small patient numbers, we found that the LF/HF ratio was numerically increased in patients with severe SDB compared to patients without SDB.

Another measure of autonomic activity is baroreflex sensitivity, which assesses the function of the baroreflex. The baroreflex is a regulatory mechanism that is crucial in the homeostasis of blood pressure by influencing the peripheral vascular tone and CO [19]. An impairment in baroreceptor reflex function is a prognostic marker in HF and has been suggested to be involved in the cardiovascular complications of OSA and CSR, increasing the risk of cardiovascular morbidity, as well as overall mortality [17,22,34,49]. Both patients with OSA and CSR show a reduced baroreflex sensitivity [44,50]. Assessing differences in HF patients, we observed a trend towards a lower BRS in patients with OSAS and CSR than without SDB. A decrease in BRS was also seen in a subgroup analysis of HF patients with severe SDB and no SDB, without, however, reaching statistical significance. These findings of other studies assessing baroreflex function in OSA patients state that baroreceptor function decreases in patients with severe OSA compared to patients with mild OSA or healthy controls [51,52]. Moreover, a study by Blomster et al. showed no reduction in baroreceptor function in patients with mild OSA versus healthy controls [53]. Similarly, in CSR patients, impairment of baroreceptor function was demonstrated only in patients with AHI ≥ 30 [49]. The lack of differences regarding BRS between groups within our whole study population may, thus, be explained by the fact that our study population consisted predominately of patients with mild SDB. Similar to our findings, Yoshihisa and co-workers investigated heart rate turbulence (HRT), another marker of the baroreceptor response in patients with CHF and SDB. The authors performed polygraphy rather than polysomnography in their study but were similarly able to confirm a blunted baroreceptor response in those with severe SDB, with a modest—but statistically significant—relationship between the severity of SDB and the baroreceptor function [54]. Our study extends these observations, as we not only provide more comprehensive hemodynamic measurements but have further stratified according to different types of SDB and report long-term survival data in our cohort of CHF patients. The treatment of OSA and CSR in patients with HF by continuous positive airway pressure was found to improve HR, systolic BP, and BRS within a month in a study by Ruttanaumpawan et al. [55]. In our study, while 37.8% of patients with SDB received treatment with positive airway pressure therapy, this was not associated with improved overall survival. Nevertheless, we found that HF patients with moderate and severe SDB had a shorter median survival of 7.9 (5.0; 13.1) months, which was, however, dependent upon age. In contrast, to a study by Hetland et al., who report a higher mortality for patients with CSR than with OSA, our study could not confirm these results [56].

The current study suffers from a number of limitations that need to be addressed. First and foremost, the number of participants in our study, particularly in the subgroups of patients with different types of SDB, was relatively small and, thus, potentially subject to both type 1 and type II errors. Furthermore, the current findings were not further corrected for drug treatment or other comorbidities, such as atrial fibrillation, which may have had an impact on some of the findings, particularly with regard to survival data. However, we would like to point out that patients in this study were selected on the basis of stringent in- and exclusion criteria and received optimized medical management for their underlying congestive heart failure at specialized clinics. Thus, this study cohort consists only of patients who underwent maximized medical management and, therefore, represent a valid cohort of “end-stage” heart failure patients. In contrast, this may prevent the generalizability of our findings to larger patient cohorts with CHF. We further need to acknowledge that our assessments of hemodynamic function and cardiovascular autonomous function were obtained during a short recording, rather than during a 24 h ECC long-term recording. On the other hand, we can rely on SDB being diagnosed by the gold-standard method of polysomnography and provide comprehensive, non-invasive hemodynamic measurements taken by validated testing methods.

## 5. Conclusions

In conclusion, the diagnosis of SDB per se did not appear to be associated with lower baroreceptor reflex sensitivity or a higher heart rate variability in this report. Nevertheless, there was a signal of impaired spontaneous baroreceptor function in chronic HF patients with severe SDB that deserves further attention. Recognizing the importance of an impaired baroreceptor function in patients with severe SDB and CHF may help potentially identify those at higher risk of major cardiovascular events. In addition to treating comorbid sleep-disordered breathing in these patients, these findings may indicate the potential need for further cardiovascular treatment escalation and/or closer follow-up assessments.

## Figures and Tables

**Figure 1 jcm-13-07219-f001:**
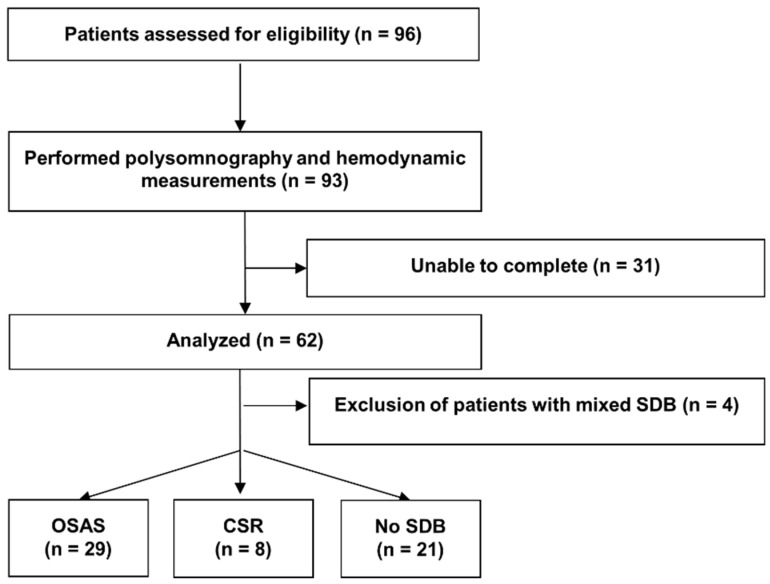
Patients flow. SDB = sleep-disordered breathing, OSAS = obstructive sleep apnea syndrome, CSR = Cheyne-Stokes respiration.

**Figure 2 jcm-13-07219-f002:**
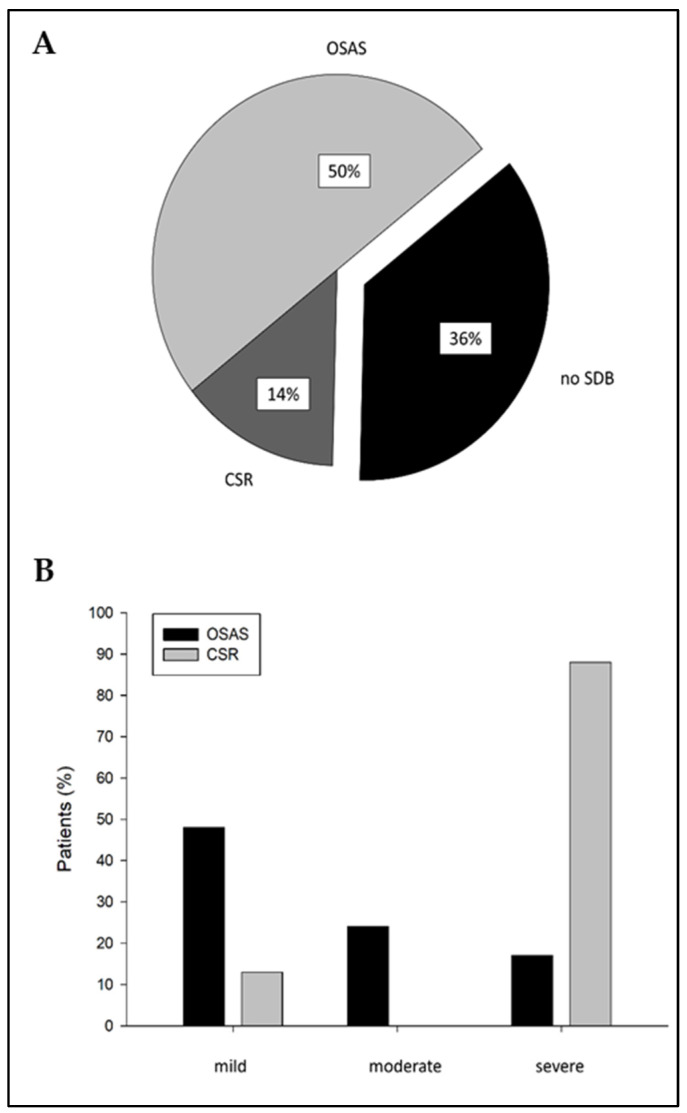
(**A**) Distribution of sleep-disordered breathing in chronic HF patients (*n* = 58) categorized in OSAS, CSR, and patients without SDB. (**B**) Distribution of SDB severity categorized in OSAS and CSR; SDB = sleep-disordered breathing, OSAS = obstructive sleep apnea syndrome, CSR = Cheyne–Stokes respiration.

**Figure 3 jcm-13-07219-f003:**
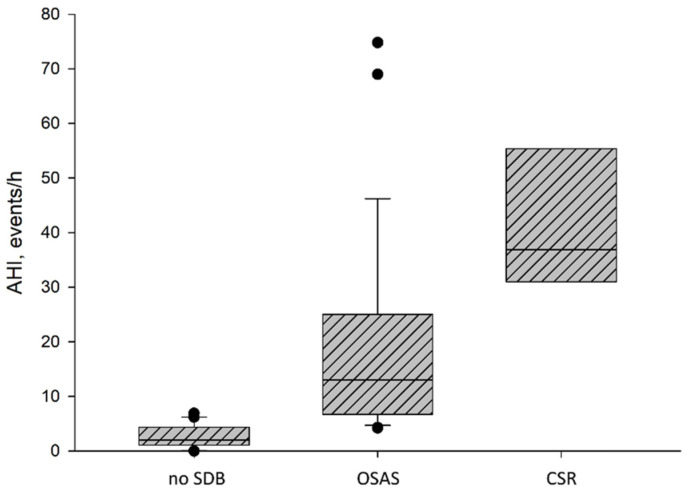
Distribution of AHI by diagnosis of OSAS, CSR, and no SDB; AHI = apnea–hypopnea index. SDB = sleep-disordered breathing, OSAS = obstructive sleep apnea syndrome, CSR = Cheyne–Stokes respiration.

**Figure 4 jcm-13-07219-f004:**
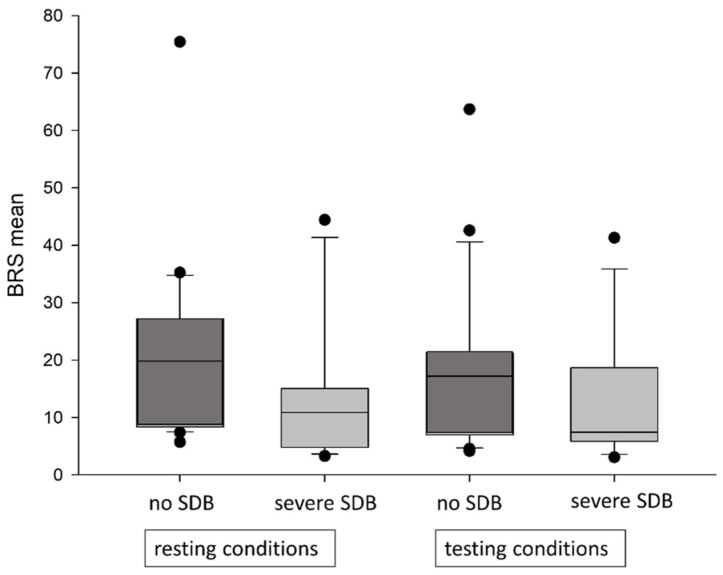
Mean BRS under resting conditions and testing conditions in patients with severe SDB and no SDB. BRS = baroreflex sensitivity, SDB = sleep-disordered breathing.

**Figure 5 jcm-13-07219-f005:**
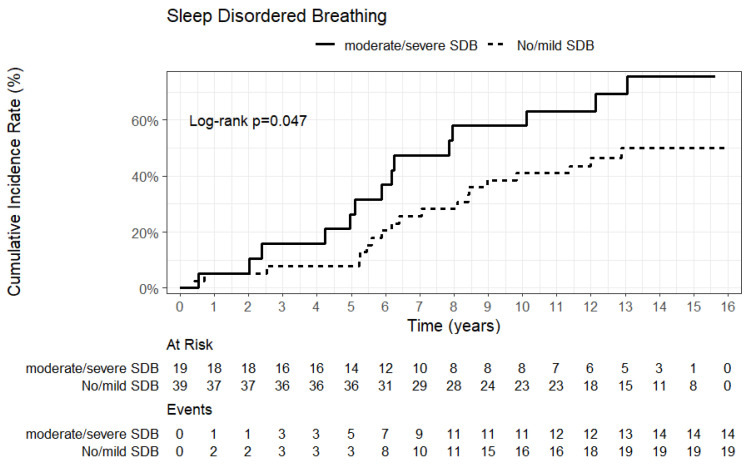
Cumulative incidence rate and numbers at events and risk in patients with no-to-mild vs. moderate-to-severe SDB. SDB = sleep-disordered breathing.

**Table 1 jcm-13-07219-t001:** Results of the overnight sleep examination for the study population.

	OSAS (*n* = 29)	CSR (*n* = 8)	No SDB (*n* = 21)	*p*-Value
Sleep study duration, h *	7.2 (6.6–8.2)	7.1 (6.6–8.1)	7.2 (6.5–7.8) ^a^	0.920
AHI, events/h *	13 (7–22)	37 (31–51)	2 (1–4)	0.000
REM-AHI, events/h *	10 (1–23) ^b^	24 (8–35)	2 (0–6) ^c^	0.009
Supine-AHI, events/h *	24 (14–46)	61 (41–71)	3 (1–6)	0.000
Arousals *	13 (10–21)	21 (11–34)	8 (6–10) ^a^	0.001
Min SaO_2_, % *	85 (80–86)	82 (80–85)	89 (88–92)	0.000
Mean SaO_2_, % *	94 (93–95)	94 (93–94)	95 (94–96)	0.003
Sleep efficacy *	72 (55–80)	78 (60–87)	78 (61–85) ^a^	0.661
paCO_2_	40 (±4) ^b^	36 (±4) ^d^	39 (±4) ^b^	0.038

Data are presented as mean (standard deviation) and compared using the one-way ANOVA unless indicated by * presented as medians with IQR and compared using the Kruskal–Wallis test. AHI = apnea–hypopnea index. SaO_2_ = oxygen saturation of arterial blood. paCO_2_ = partial pressure of carbon dioxide. ^a^ *n* = 20, ^b^ *n* = 28, ^c^ *n* = 19, ^d^ *n* = 7.

**Table 2 jcm-13-07219-t002:** Clinical characteristics of patients classified into OSAS, CSR, and no SDB.

	OSAS (*n* = 29)	CSR (*n* = 8)	No SDB (*n* = 21)	*p*-Value
Gender (F/M)	9/20	0/8	8/13	0.130
Age, years	63 (±11)	64 (±9)	59 (±12)	0.447
BMI, kg/m^2^	30 (±6)	30 (±4)	28 (±5)	0.336
Neck-to-waist ratio	0.39 (±0.03) ^a^	0.39 (±0.017)	0.39 (±0.27)	0.763
Smoker, *n* (%)	12 (41.4%)	6 (75.0%)	5 (23.8%)	**0.040**
Smoking, pack years *	12 (0–40)	19 (8–45)	20 (0–35)	0.104
NYHA 1/2/3	12/15/2	2/5/1	6/13/2	0.549
6 min walk test	348(±93) ^b^	417(±73) ^c^	367(±125) ^d^	0.323
Hypertension, *n* (%)	21 (72.4%)	4 (50.0%)	10 (47.6%)	0.177
Diabetes mellitus, *n* (%)	11 (37.9%)	1 (12.5%)	8 (38.1%)	0.394
Hyperlipidemia, *n* (%)	12 (41.4%)	5 (62.5%)	9 (45.0%) ^e^	0.582
MLHFQ *	20 (9–43) ^f^	7 (3–34) ^g^	21 (3–33) ^h^	0.392
ESS *	6 (4–10)	7 (2–10) ^g^	5 (3–9)	0.734
PSQI	7 (±3) ^i^	6 (±3) ^g^	6 (±4)	0.805

Data are presented as mean (standard deviation) and compared using the one-way ANOVA unless indicated by * presented as medians with IQR and compared using the Kruskal–Wallis test. Bold values show significant results. BMI = body mass index. SDB = sleep-disordered breathing, OSAS = obstructive sleep apnea syndrome, CSR = Cheyne–Stokes respiration, NYHA = New York Heart Association Classification, MLHFQ = Minnesota Living with Heart Failure Questionnaire (max. 105 points), ESS = Epworth Sleepiness Scale (max. 24 points). PSQI = Pittsburgh Sleep Quality Index (max. 21 points). ^a^ *n* =27, ^b^ *n* = 23, ^c^ *n* = 7, ^d^ *n* = 18, ^e^ *n* = 20, ^f^ *n* = 25, ^g^ *n* = 7, ^h^ *n* = 19, ^i^ *n* = 28.

**Table 3 jcm-13-07219-t003:** Hemodynamic and heart rate measurement at rest and under mental stress.

	OSAS (*n* = 29)	CSR (*n* = 8)	No SDB (*n* = 21)	*p*-Value
**Rest:**				
HR rest, bpm *	67 (58–69)	63 (56–70)	64 (61–69)	0.908
BPsys rest, mmHg	118 (±24)	120 (±14)	124 (±16)	0.573
BPdia rest, mmHg	75 (±16)	78 (±9)	76 (±15)	0.847
SV rest, mL	73 (±15)	74 (±16)	69 (±11)	0.515
SI rest, mL/m^2^	36 (±7)	36 (±9)	35 (±6)	0.631
CO rest, L/min	4.7 (±0.8)	4.7 (±0.7)	4.5 (±0.7)	0.616
CI rest, L/min/m^2^	2.3 (±0.4)	2.3 (±0.4)	2.2 (±0.4)	0.628
LF/HF-RRI rest *	0.9 (0.4–1.4)	2 (0.9–3.4)	0.7 (0.5–1.6)	0.112
LF/HF rest *	0.6 (0.4–0.9)	1.4 (0.6–1.9)	0.5 (0.3–0.8)	0.101
TRP rest	1491 (±428)	1341 (±605)	1620 (±365)	0.281
**Mental stress:**				
HR mental stress, bpm *	69 (64–75)	65 (58–69)	68 (62–74)	0.450
BP systolic mental stress, mmHg *	130 (111–149)	129 (119–144)	127 (119–141)	0.988
BP diastolic mental stress, mmHg *	81 (75–92)	82 (75–96)	83 (72–86)	0.875
SV mental stress, mL	72 (±15)	80 (±15)	70 (±12)	0.185
SI mental stress, mL/m^2^	36 (±7)	39 (±9)	35 (±6)	0.391
CO mental stress, L/min	5.0 (±0.9)	5.4 (±1)	4.8 (±0.9)	0.299
CI mental stress, L/min/m^2^	2.5 (±0.4)	2.6 (±0.6)	2.4 (±0.5)	0.623
LF/HF-RRI mental stress *	0.6 (0.4–1.8)	1.8 (0.7–2)	0.9 (0.4–1.7)	0.305
LF/HF mental stress *	0.7 (0.4–1.0)	0.8 (0.5–1.5)	0.6 (0.4–0.9)	0.576
TRP mental stress	1536 (±427)	1503 (±297)	1646 (±418)	0.580

Data are presented as mean (standard deviation) and compared using the one-way ANOVA unless indicated by * presented as medians with IQR and compared using the Kruskal–Wallis test. SDB = sleep-disordered breathing, OSAS = obstructive sleep apnea syndrome, CSR = Cheyne–Stokes respiration, HR = heart rate. BP = blood pressure. SV = stroke volume. SI = stroke index. LF/HF = low frequency/high frequency. TRP = total peripheral resistance.

**Table 4 jcm-13-07219-t004:** Heart rate variability in patients with severe SDB and patients without SDB.

	Severe SDB (*n* = 12)	No SDB (*n* = 21)	*p*-Value
**Rest:**			
LF/HF-RRI rest	1.4 (0.6–2.6)	0.8 (0.5–1.4)	0.321
LF/HF rest	0.9 (0.4–1.8)	0.5 (0.3–0.8))	0.184
**Mental stress:**			
LF/HF-RRI mental stress	1.2 (0.5–2)	0.9 (0.4–1.7)	0.489
LF/HF mental stress	0.7 (0.5–1.2)	0.6 (0.4–0.9)	0.501

Data are presented as medians with IQR and compared using the Kruskal–Wallis test. SDB = sleep-disordered breathing, LF/HF = low frequency/high frequency.

**Table 5 jcm-13-07219-t005:** Baroreceptor reflex sensitivity results for the study population.

	OSAS (*n* = 29)	CSR (*n* = 8)	No SDB (*n* = 21)	*p*-Value
BRS slope mean, rest	14 (8–34)	9 (7–14)	20 (8–26)	0.157
BRS slope mean, mental test	12 (6–24)	8 (5–12)	17 (8–21)	0.140
ΔBRS mean	1.5 (−3.5–5.6)	1.2 (−0.7–4.9)	3.2 (−0.1–6.7)	0.596

Data are presented as medians with IQR and compared using the Kruskal–Wallis test. SDB = sleep-disordered breathing, OSAS = obstructive sleep apnea syndrome, CSR = Cheyne–Stokes respiration, BRS = baroreflex sensitivity.

## Data Availability

The data presented in this study are available on reasonable request from the corresponding author.

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
