# Peer review of "Sleep-Disordered Breathing in Patients with Chronic Heart Failure and Its Implications on Real-Time Hemodynamic Regulation, Baroreceptor Reflex Sensitivity, and Survival"

_jcm, 2024, doi:10.3390/jcm13237219_

Round 1
Reviewer 1 Report
Comments and Suggestions for Authors
This paper reports the analysis of derived cardiovascular variables, BRS, sleep quality and excessive daytime sleepiness in patients with CHF undergoing PSG, followed for over 10 years. The sample only included 58 patients, 50% of them showed OSA, only 8 patients had CSR, and the remaining patients were free of SDB. When considered according to SDB severity, the majority of patients with severe SDB had CSR. The variables analyzed did not differ among the 3 groups at baseline, only BRS showed a trend for lower values in patients with severe SDB. Moreover, differences in survival disappeared when results were corrected for age. The authors conclude that SDB did not affect HRV or BRS (but the latter should be confirmed) in patients with CHF.
Comments:
The sample is small and not balanced between OSA and CSR groups. This may affect significance of the results.
The follow-up is not mentioned, how were the data collected? DId you also collect data on CV morbidity?
A major methodological issue not mentioned is: did you exclude patients with atril fibrillation? If not, ple
Pharmacological treatment is not considered as a major possible cause of lack of differences among groups of patients, were analyses corrected for drug tratment? please discuss.
I was surprised by the lack of differences in ABG among groups. But again the numbers were small and not all patients were investigated.
Mean BRS slope showed a definite trend, did you correlate the slope with AHI? You may wish to show OSA and CSR patients with different symbols while considering all patients in a single simple regression.
Author Response
"Please see the attachment."

Reviewer 2 Report
Comments and Suggestions for Authors
Sleep disordered breathing is a really important topic. In this regard, the relevance of the manuscript is beyond doubt. The manuscript is well structured. However, some clinical clarifications are required. The authors need to indicate whether the patients took medications that affect sleep, what was the sleep-wake regimen of the studied groups of patients? Were the bedtime, wake-up time, and total number of hours spent on sleep the same? Did the patients have daytime sleep? It would also be appropriate for the authors to indicate the analysis of the polysomnography results, the number of apneas during the studied period of time, the amount of time spent on each sleep phase, and whether this indicator differed in the observation groups. The authors indicate the mortality rate in the groups. However, the manuscript does not contain information on the causes of death, this information needs to be added and analyzed. In the conclusion section, the authors need to provide information on the possible application of the results they obtained in practical healthcare; this will significantly increase the interest of readers and the value of the manuscript.
Author Response
"Please see the attachment."

Reviewer 3 Report
Comments and Suggestions for Authors
The authors conducted an observational cohort study to investigate implications of sleep disordered breathing (SDB) on hemodynamic regulation and autonomic activity in chronic HF patients. They report that they found no differences in real-time assessment of hemodynamic regulation, heart rate variability or baroreceptor reflex function between patients with OSAS, CSR, and patients without SDB. Subgroup analysis of BRS and HRV in patients with severe SDB (AHI>30/h) and without SDB (AHI<5) revealed numerically reduced BRS and increased LF/HF-RRI values under resting conditions as well as during mental testing in patients with severe SDB. Patients with moderate to severe SDB had a shorter overall survival, which was, however, dependent on age. Suggestions:
1) Please present the manuscript as per STROBE reporting guidelines and STROBE checklist.
2) Were there any power calculations?
3) Please quantify missing data. How was it handled?
4) Was there any adjustment for multiple comparisons testing such as Bonferroni correction?
5) Instead of survival curve, consider presenting cumulative incidence curve.
6) In the limitations para, please add & discuss the potential for selections bias; misclassification of exposure; misclassification of outcome; unmeasured and residual confounding; time-varying confounding; other factors limiting generalizability of findings; potential for type I and type II error.
7) Please consider discussing and citing the following relevant study and how the findings from this study relate to it:
Yoshihisa A, Suzuki S, Takiguchi M, Shimizu T, Abe S, Sato T, Yamaki T, Sugimoto K, Kunii H, Nakazato K, Suzuki H, Saitoh S, Takeishi Y. Impact of sleep-disordered breathing on heart rate turbulence in heart failure patients. PLoS One. 2014 Jun 26;9(6):e101307. doi: 10.1371/journal.pone.0101307. PMID: 24968229; PMCID: PMC4072775.
Author Response
"Please see the attachment."
